# Improving Data-Efficiency and Robustness of Medical Imaging Segmentation Using Inpainting-Based Self-Supervised Learning

**DOI:** 10.3390/bioengineering10020207

**Published:** 2023-02-04

**Authors:** Jeffrey Dominic, Nandita Bhaskhar, Arjun D. Desai, Andrew Schmidt, Elka Rubin, Beliz Gunel, Garry E. Gold, Brian A. Hargreaves, Leon Lenchik, Robert Boutin, Akshay S. Chaudhari

**Affiliations:** 1Department of Radiology, Stanford University, Stanford, CA 94305, USA; 2Department of Electrical Engineering, Stanford University, Stanford, CA 94305, USA; 3Department of Bioengineering, Stanford University, Stanford, CA 94305, USA; 4Department of Radiology, Wake Forest University School of Medicine, Winston-Salem, NC 27101, USA; 5Department of Biomedical Data Science, Stanford University, Stanford, CA 94305, USA; 6Stanford Cardiovascular Institute, Stanford University, Stanford, CA 94305, USA

**Keywords:** self-supervised learning, MRI, CT, segmentation, machine learning, deep learning

## Abstract

We systematically evaluate the training methodology and efficacy of two inpainting-based pretext tasks of context prediction and context restoration for medical image segmentation using self-supervised learning (SSL). Multiple versions of self-supervised U-Net models were trained to segment MRI and CT datasets, each using a different combination of design choices and pretext tasks to determine the effect of these design choices on segmentation performance. The optimal design choices were used to train SSL models that were then compared with baseline supervised models for computing clinically-relevant metrics in label-limited scenarios. We observed that SSL pretraining with context restoration using 32 × 32 patches and Poission-disc sampling, transferring only the pretrained encoder weights, and fine-tuning immediately with an initial learning rate of 1 × 10${}^{-3}$ provided the most benefit over supervised learning for MRI and CT tissue segmentation accuracy (*p* < 0.001). For both datasets and most label-limited scenarios, scaling the size of unlabeled pretraining data resulted in improved segmentation performance. SSL models pretrained with this amount of data outperformed baseline supervised models in the computation of clinically-relevant metrics, especially when the performance of supervised learning was low. Our results demonstrate that SSL pretraining using inpainting-based pretext tasks can help increase the robustness of models in label-limited scenarios and reduce worst-case errors that occur with supervised learning.

## 1. Introduction

Segmentation is an essential task in medical imaging that is common across different imaging modalities and fields such as cardiac, abdominal, musculoskeletal, and lung imaging, amongst others [1,2,3,4]. Deep learning (DL) has enabled high performance on these challenges, but the power-law relationship between algorithmic performance and the amount of high-quality labeled training data fundamentally limits robustness and widespread use [5].

Recent advances in self-supervised learning (SSL) provide an opportunity to reduce the annotation burden for deep learning models [6]. In SSL, a model is first pretrained on a “pretext” task, during which unlabeled images are perturbed and the model is trained to predict or correct the perturbations. The model is then fine-tuned for downstream tasks. Previous works have shown that such models can achieve high performance even when fine-tuned on only a small labeled training set [7,8,9]. While most SSL models in computer vision have been used for the downstream task of image classification, segmentation comparatively remains an under-explored task [10].

In this work, we systematically evaluate the efficacy of SSL for medical image segmentation across two domains—MRI and CT. We investigate “context prediction” [7] and “context restoration” [8], two well-known and easy-to-implement archetypes of restoration-based pretext tasks that produce image-level representations during pretraining for eventual fine-tuning. Context prediction sets pixel values in random image patches to zero, while context restoration randomly swaps pairs of image patches within an image while maintaining the distribution of pixel values (Figure 1). For both tasks, the model needs to recover the original image given the corrupted image, a process we refer to as “inpainting”. We consider these two tasks because they maintain same input-output sizes, akin to segmentation. We hypothesize that such pretext tasks allow construction of useful, image-level representations that are more suitable for downstream segmentation.

While context prediction and context restoration have been proposed before, the effects of the large space of design choices for these two pretext tasks, such as patch sizes for image corruption and learning rates for transfer learning, are unexplored. In addition, prior works exploring SSL for medical image segmentation have primarily focused on the accuracy of segmentation using metrics such as Dice scores [8,11], but have not investigated if SSL can improve clinically-relevant metrics, such as T2 relaxation times for musculoskeletal MRI scans and mean Hounsfield Unit (HU) values for CT scans. These metrics can provide biomarkers of biochemical changes in tissue structure prior to the onset of gross morphological changes [12,13]. Furthermore, within the context of empirical data scaling laws in DL, past SSL works have rarely explored benefits of increasing the number of unlabeled images during pretraining [14]. Characterizing the efficiency of SSL methods with unlabeled data can lead to more informed decisions regarding data collection, an important practical consideration for medical image segmentation. In this work, we address the above gaps by (1) investigating how different design choices in SSL implementation affect the quality of the pretrained model, (2) calculating how varying unlabeled data extents affects SSL performance for downstream segmentation, (3) quantifying our results using clinically-relevant metrics to investigate if SSL can outperform supervised learning in label-limited scenarios, (4) evaluating where SSL can improve performance, across different extents of labeled training data availability, and (5) providing detailed analyses, recommendations, and open-sourcing our code to build optimal SSL models for medical image segmentation (code available at https://github.com/ad12/MedSegPy).

## 2. Materials and Methods

### 2.1. Datasets

#### 2.1.1. MRI Dataset

We used 155 labeled knee 3D MRI volumes (around 160 slices per volume) from the SKM-TEA dataset [15] and 86 unlabeled volumes (around 160 to 180 slices per volume), each with slice dimensions of 512 × 512 (other scan parameters in [15]). All volumes were acquired using a 5-min 3D quantitative double-echo in steady-state (qDESS) sequence, which has been used for determining morphological and quantitative osteoarthritis biomarkers and for routine diagnostic knee MRI [16,17,18,19]. The labeled volumes included manual segmentations for the femoral, tibial, and patellar cartilages, and the meniscus. The labeled volumes were split into 86 volumes for training, 33 for validation, and 36 for testing, following the splits prescribed in [15]. The 86 training volumes were further split into additional subsets, consisting of 50% (43 volumes), 25% (22 volumes), 10% (9 volumes), and 5% (5 volumes) training data, to represent label-limited scenarios. All scans in smaller subsets were included in larger subsets.

#### 2.1.2. CT Dataset

The 2D CT dataset consisted of 886 labeled and 7799 unlabeled abdominal CT slices at the L3 vertebral level. The unlabeled images were used in a prior study exploring the impact of body composition on cardiovascular outcomes [20]. The labeled slices included manual segmentations for subcutaneous, visceral, and intramuscular adipose tissue and muscle. These labeled slices were split into 709 slices for training, 133 for validation, and 44 for testing. The training set was split in a similar manner as the MRI volumes into 4 additional subsets of 50% (354 slices), 25% (177 slices), 10% (71 slices), and 5% (35 slices) training data. No metadata from the dataset were used in any models.

### 2.2. Data Preprocessing

All models segmented 2D slices for MRI and CT images. Each CT image was preprocessed at different windows and levels (*W/L*) of HU to emphasize different image contrasts, resulting in three-channel images: soft-tissue (*W/L* = 400/50), bone (*W/L* = 1800/40), and a custom setting (*W/L* = 500/50). All images were normalized to have zero mean and unit standard deviation, with MR images normalized by volume and CT images normalized per channel.

### 2.3. Model Architecture and Optimization

2D U-Net models [21] with Group Normalization [22], weight standardization [23], and He random weight initializations [24] were used for inpainting and segmentation (Figure 2). Both inpainting and segmentation used identical U-Nets, except for the final convolutional layer, which we refer to as the “post-processing” layer. For inpainting, the post-processing layer produced an output image with the same number of channels as the input image, whereas for segmentation, it produced a 4-channel image for the four segmentation classes in each dataset.

We used L2 norm loss for inpainting and Dice loss, aggregated over mini-batches per segmentation class, for segmentation. All training was performed with early stopping and the ADAM optimizer [25] ($\beta_{1}$ = 0.99 and $\beta_{2}$ = 0.995) with a batch size of 9 on an NVIDIA 2080Ti GPU. Additional details are in Section A.1.

### 2.4. Image Corruption for Pretext Tasks

We incorporated random block selection to select the square image patches to corrupt during pretraining. To ensure the amount of corruption per image was fixed and did not affect later comparison, the patches for each image were iteratively selected and corrupted until 1/4 of the total image area was corrupted.

For context prediction, we selected and set random patches of dimensions *K* × *K* to zero in an iterative manner until the number of pixels set to zero equaled or exceeded 1/4 of the total image area. For context restoration, randomly selected pairs of non-overlapping *K* × *K* image patches were swapped in an iterative manner until the number of corrupted pixels equaled or exceeded 1/4 of the total image area. We refer to the result of both methods as “masks”. The context prediction binary mask specified which pixels were zero and the context restoration mask was a list of patch pairs to be swapped. When pretraining with multi-channel CT images, the locations of the patch corruptions were identical across channels to avoid shortcut learning [26]. Example image corruptions are shown in Figure 1.

To train the model to inpaint any arbitrarily corrupted image region without memorization of image content, we sampled a random mask every iteration for all images. For computational efficiency, we precomputed 100 random masks before training. We further randomly rotated the masks by either 0, 90, 180, or 270° counter-clockwise to increase the effective number of masks used during training to 400.

### 2.5. Design Choices for SSL Implementation

Design choices for inpainting-based SSL segmentation revolving around pretraining task implementations [7,8] and transfer learning [27,28,29] have not been systematically compared. To overcome these shortcomings, we explored the following questions:Which pretrained weights should be transferred for fine-tuning?How should the transferred pretrained weights be fine-tuned?What should be the initial learning rate when fine-tuning?What patch size should be used when corrupting images for inpainting?How should the locations of the patches be sampled when corrupting images for inpainting?

#### 2.5.1. Design Choices for Transfer Learning (#1–3)

For design choice #1 (which pretrained weights to transfer), we compared transferring only the U-Net encoder weights [7] with transferring both the encoder and decoder weights [8].

For design choice #2, we compare (i) fine-tuning all pretrained weights immediately after transferring [27,28], and (ii) freezing pretrained weights after transferring and training until convergence, then subsequently unfreezing pretrained weights and training all weights until convergence [29,30].

For design choice #3, we selected four initial learning rates: 1 × 10${}^{-2}$, 1 × 10${}^{-3}$, 1 × 10${}^{-4}$, and 1 × 10${}^{-5}$, to evaluate whether pretrained features are distorted with larger learning rates.

To compare different combinations of these three design choices, we performed a grid search and defined the best combination to be the one with the best segmentation performance on the MRI test set when trained with the MRI training subset with 5% training data. More details are in Section B.1.

#### 2.5.2. Design Choices for Pretraining (#4–5)

For design choice #4, we compare patch sizes of 64 × 64, 32 × 32, 16 × 16, and 8 × 8 (Figure 1). For design choice #5, we compare two sampling methods: (i) fully-random sampling where the location of each patch was selected at random and constrained to lie completely within the image [7,8], and (ii) Poisson-disc sampling that enforces the centers of all *K* × *K* patches to lie at least $K\sqrt{2}$ pixels away from each other to prevent overlapping patches [31]. To compare different combinations of design choices #4 and #5 and the two pretext tasks, we performed a grid search by training a model for each combination five times, each time using one of the five training data subsets, for both datasets. We also trained a fully-supervised model for each dataset and training data subset for a baseline comparison. All models were fine-tuned in an identical manner with the same random seed after pretraining, using the best combination of design choices #1–3. All inpainting models were compared by computing the L2 norm of the generated inpainted images. When computing the L2 norm value for each three-channel CT image, the L2 norm value was computed per channel and averaged across all channels. All segmentation models were compared by computing the Dice coefficient for each segmentation class in the test set, averaged across all available volumes/slices.

#### 2.5.3. Optimal Pretraining Evaluation

We defined the optimal pretraining strategy as the strategy that provided the most benefit over supervised learning, across image modalities and training data extents, in the experiment described in Section 2.5.2.

For each baseline (fully-supervised model) and SSL model trained in the experiment using 50%, 25%, 10%, and 5% training data, we computed class-averaged Dice scores for every test volume/slice in the MRI and CT datasets. For each pretraining strategy and dataset, we compared whether the set of Dice scores of the corresponding SSL models were significantly higher than that of the respective fully-supervised models using one-sided Wilcoxon signed-rank tests. As a heuristic, the pretraining strategies were sorted by their associated *p*-values and the pretraining strategy that appeared in the top three for both the MRI and CT datasets was selected as the optimal pretraining strategy. We defined the optimally trained model for each dataset as the SSL model that was pretrained with this optimal pretraining strategy and fine-tuned for segmentation using the best combination of design choices #1-3.

### 2.6. Impact of Extent of Unlabeled Data

To measure the effect of the number of pretraining images on downstream segmentation performance, the optimally trained model was pretrained with the standard training set as well as two supersets of the training set containing additional unlabeled imaging data. We refer to the standard training set as 100% pretraining data (86 volumes for MRI and 709 slices for CT). For the MRI dataset, the second and third sets consisted of 150% (129 volumes) and 200% (172 volumes) pretraining data, respectively. For the CT dataset, the second and third sets consisted of 650% (4608 slices) and 1200% (8508 slices) pretraining data, respectively. After pretraining, all the pretrained models were fine-tuned with the five subsets of labeled training data and a Dice score was computed for each fine-tuned model, averaged across all segmentation classes and all volumes/slices in the test set. To quantify the relationship between Dice score and the amount of pretraining data for each subset of labeled training data, a curve of best fit was found using non-linear least squares. The Residual Standard Error, defined as $\frac{\sum_{i=1}^{n}{\left(y_{i}-\hat{y_{i}}\right)}^{2}}{n-2}$, was computed to quantify how well the curve of best fit fits the data.

For MRI and CT, the pretraining dataset that led to the best average Dice score across the extents of labeled training data was chosen for further experiments.

### 2.7. Comparing SSL and Fully-Supervised Learning

We compared baseline fully-supervised models and the optimally trained models pretrained with the chosen pretraining dataset from the experiment described in Section 2.6. For each training data subset, models were evaluated using two clinically-relevant metrics for determining cartilage, muscle, and adipose tissue health status. For MRI, we computed mean T2 relaxation time per tissue and tissue volume [32]. For CT, we computed cross-sectional area and mean HU value per tissue. We calculated their percentage errors by comparing them to values derived from using ground truth segmentations to compute the metrics.

To determine which images benefit maximally with SSL, we compared and visualized the percentage error in the clinically-relevant metrics between supervised learning and SSL. For both supervised learning and SSL, the percentage error for each test image was averaged over all classes and label-limited scenarios.

### 2.8. Statistical Analysis

All statistical comparisons were computed using one-sided Wilcoxon signed-rank tests. All statistical analyses were performed using the SciPy (v1.5.2) library [33], with Type-1 α=0.05.

## 3. Results

The subject demographics of all labeled and unlabeled volumes/slices are shown in Table 1.

### 3.1. Design Choices for Transfer Learning

We observed that all pretrained model variants had high performance when first fine-tuned with an initial learning rate of 1 × 10${}^{-3}$ and then fine-tuned a second time with an initial learning rate of 1 × 10${}^{-4}$. Transferring pretrained encoder weights only and fine-tuning once immediately with an initial learning rate of 1 × 10${}^{-3}$ achieved similar performance, with the added benefit of reduced training time. Consequently, we used these as the best combination of the three design choices for transfer learning. Additional details are in Section B.2.

### 3.2. Design Choices for Pretraining

The L2 norm consistently decreased as a function of patch size for all combinations of pretext tasks (context prediction and context restoration) and sampling methods (random and Poisson-disc) (Table 2). Furthermore, L2 norms for Poisson-disc sampling were significantly lower than those for random sampling (*p* < 0.05).

Dice scores for fully-supervised baselines ranged from 0.67–0.88 across subsets of training data for MR images. Downstream segmentation performance for the MRI dataset was similar for all combinations of pretext task, patch size, and sampling method (Figure 3). All SSL models matched (within 0.01) or outperformed the fully-supervised model in low-label regimes with 25% training data or less for the femoral cartilage, patellar cartilage, and meniscus, and had comparable performance for higher data extents. For the tibial cartilage, all SSL models outperformed the fully-supervised model when trained on 5% training data and had comparable performance for higher data extents. The difference in Dice score between each self-supervised model and the fully-supervised model generally increased as the amount of labeled training data decreased. SSL pretraining also enabled some models to outperform the fully-supervised model trained with 100% training data in patellar cartilage segmentation.

Dice scores for fully-supervised baselines were consistently higher for CT images than for MR images, with the exception of intramuscular adipose tissue. Unlike with the MRI dataset, downstream SSL segmentation for CT in low-label regimes depended on the pretext task and the patch size used during pretraining (Figure 4). Models pretrained with larger patch sizes (64 × 64; 32 × 32) often outperformed those pretrained with smaller patch sizes (16 × 16; 8 × 8) for muscle, visceral fat, and subcutaneous fat segmentation, when trained with either 5% or 10% labeled data. Furthermore, when 25% training data or less was used, models pretrained with 32 × 32 patches using context restoration almost always outperformed fully-supervised models for muscle, visceral fat, and subcutaneous fat segmentation, but rarely did so when pretrained using context prediction. For intramuscular fat, all SSL models had comparable performance with fully-supervised models in low-label regimes. For high-label regimes (over 25% labeled data), all SSL models had comparable performance with fully-supervised models for all four segmentation classes.

### 3.3. Optimal Pretraining Evaluation

The top 5 pretraining strategies for the MRI dataset and the top 3 pretraining strategies for the CT dataset led to significantly better segmentation performance compared to fully-supervised learning (*p* < 0.001) (Table 3).

For MRI, the top 5 strategies all consisted of pretraining with context restoration, with minimal differences in *p*-value based on the patch size and sampling method used. For CT, the top 5 strategies used a patch size of at least 32 × 32 during pretraining. The strategy of pretraining with context restoration, 32 × 32 patches, and Poisson-disc sampling was in the top 3 for both datasets, and was therefore selected as the optimal pretraining strategy.

### 3.4. Impact of Extent of Unlabeled Data

For both datasets and for most subsets of labeled training data used during fine-tuning (except 25% and 10% labeled training data for MRI), the optimally trained model performed significantly better in downstream segmentation when pretrained on the maximum amount of data per dataset (200% pretraining data for MRI and 1200% pretraining data for CT) than when pretrained on only the training set (*p* < 0.05) as seen in Figure 5. When 25% or 10% labeled training data was used for MRI segmentation, the optimally trained model achieved a higher mean Dice score when pretrained on 200% pretraining data, but this was not statistically significant (*p* = 0.3 for 25% labeled training data and *p* = 0.02 for 10% labeled training data).

For MRI, Dice scores almost always improved as the amount of pretraining data increased. This improvement was greatest when only 5% of the labeled training data was used for training segmentation. Improvements in segmentation performance were slightly higher for CT. For all extents of labeled training data, segmentation performances improved when the amount of pretraining data increased from 100% to 650%. There was limited improvement when the amount of pretraining data increased from 650% to 1200%. For both datasets, when 25%, 10%, or 5% of the labeled training data was used, the change in dice score as a function of the amount of pretraining data followed a power-law relationship of the form $y=ax^{k}+c$ (residual standard errors ≤ 0.005), where the value of *k* was less than 0.5.

Pretraining on the maximum amount of data enabled the optimally trained models to surpass the performance of fully-supervised models for all extents of labeled training data, in both MRI and CT. For the MRI dataset, the highest improvement over supervised learning was observed when 5% labeled training data was used. For CT, considerable improvements over supervised learning were observed when 5%, 10%, or 25% labeled training data was used.

For both the MRI and CT datasets, the best average Dice score over all extents of labeled training data occurred when the maximum possible amount of pretraining data was used (200% pretraining data for MRI and 1200% pretraining data for CT).

### 3.5. Comparing SSL and Fully-Supervised Learning

For each dataset, optimally trained models were pretrained with the maximum amount of pretraining data from Section 3.4.

For all clinical metrics, using optimally trained models generally led to lower percent errors than using fully-supervised models in regimes of 10% and 5% labeled training data (Figure 6). These differences were especially pronounced for CT tissue cross-sectional area, MRI tissue volume, and MRI mean T2 relaxation time. With 5% labeled training data for MRI, segmentations from optimally trained models more than halved the percent error for both tissue volume and mean T2 relaxation time of patellar cartilage, compared to segmentations from fully-supervised models.

With 100% or 50% labeled training data, percent errors for all clinical metrics had lower improvement when optimally trained models were used. This was observed for CT tissue cross-sectional area, CT mean HU value, and MRI T2 relaxation time, where optimally trained models had similar or slightly worse performance than fully-supervised models when 100% or 50% labeled data was available. However, for MRI tissue volume, optimally trained models almost always outperformed the fully-supervised models, even in scenarios with large amounts of labeled training data.

For both datasets, clinical metrics improved the most for the most challenging classes to segment. This included intramuscular adipose tissue for CT, where percent error decreased from around 3940% to 3600% for tissue cross-sectional area when 10% labeled training data was used, and patellar cartilage for MRI, where percent error decreased from around 30% to 12% for tissue volume when 5% labeled training data was used.

On a per-image basis, using SSL consistently matched or reduced the percent errors of supervised learning across both datasets and all clinical metrics (Figure 7). Furthermore, when using SSL, the percent error for all clinical metrics improved more for test images with larger percent errors when using supervised learning. For tissue cross-sectional area and mean HU value for CT, the improvement in SSL percent error gradually increased as the supervised percent error increased beyond 10%. The same pattern existed for MRI tissue volume as the supervised percent error increased beyond 20%. For MRI mean T2 relaxation time, the improvement in percent error when using SSL increased for most test images as the supervised percent error increased beyond 5%, but this was not as consistent as for the other clinical metrics. On average, when excluding intramuscular fat for CT, SSL decreased per-image percent errors for CT tissue cross-sectional area, CT mean HU value, MRI tissue volume, and MRI mean T2 relaxation time by 4.1, 1.9, 4.1, and 2.2%, respectively.

## 4. Discussion

In this work, we investigated several key, yet under-explored design choices associated with pretraining and transfer learning in inpainting-based SSL for tissue segmentation. We examined the effect of inpainting-based SSL on the performance of tissue segmentation in various data and label regimes for MRI and CT scans, and compared it with fully-supervised learning. We quantified performance using standard Dice scores and four clinically-relevant metrics of imaging biomarkers.

We observed that the crosstalk between the initial and fine-tuning learning rate was a design choice that most affected model performance. All model variants achieved optimal performance with an initial learning rate of 1 × 10${}^{-3}$ and a fine-tuning learning rate of 1 × 10${}^{-4}$ (Figure A1). This suggests the need for not perturbing the pretrained representations from the pretext task with a large learning rate. Moreover, although freezing and then fine-tuning the transferred weights provided an improvement over fine-tuning immediately for this learning rate combination (Figure A1), the improvement was very small. This result matches the findings of Kumar et al. [30], where the performance of linear probing (freezing) and then fine-tuning only slightly improved the performance of fine-tuning immediately after transferring. Additional details are provided in Section B.3.

Here, we suggest some best practices for inpainting-based SSL for medical imaging segmentation tasks. We observed that downstream segmentation performance for MRI was similar for all combinations of pretext tasks, patch sizes, and sampling techniques. This observation remained consistent despite significant differences in the L2 norms of the inpainted images. While decreasing patch sizes and sampling patch locations via Poisson-disc sampling to ensure non-overlapping patches both resulted in significantly lower L2 norms, they did not improve downstream segmentation performance. These observations suggest a discordance between learning semantically meaningful representations and the accuracy of the pretext task metric. Thus, simply performing good enough pretraining may be more important than optimizing pretext task performance.

For both MRI and CT, segmentation performance usually increased in proportion to the amount of pretraining data. The highest improvements over supervised learning were observed in the context of very low labeled data regimes of 5–25% labeled data. These empirical observations across both MRI and CT demonstrate that pretraining with large enough datasets improves performance compared to only supervised training, especially when the amount of available training data is limited.

Similar to supervised learning, improvements in SSL Dice scores tended to follow a power-law relationship of the form $y=ax^{k}+c$ as the size of the unlabeled corpora increased [5]. The observations that the value of *k* was less than 0.5 when 25%, 10%, or 5% labeled data was used for either dataset and pretraining on 650% and 1200% CT pretraining data led to similar improvements over supervised learning suggest a limit exists where the learning capacity of a model saturates and additional unlabeled data may not improve downstream performance. A good practice for future segmentation studies may be to create Figure 5 to evaluate the trade-off between the challenges of annotating more images and acquiring more unlabeled images.

Compared to fully-supervised models, optimally trained models generally led to more accurate values for all clinical metrics in label-limited scenarios. We also observed that clinical metrics improved the most with SSL for tissue classes that had the highest percent error with fully-supervised learning—intramuscular adipose tissue in CT and patellar cartilage in MRI. This observation, combined with the Dice score improvement in low labeled data regimes, suggests that SSL may be most efficacious when the performance of the baseline fully-supervised model is low.

A similar pattern was observed on a per test image basis. For all clinical metrics, the improvement in percent error when using optimally trained models was greater for test images that performed poorly when using fully-supervised models. This suggests that SSL pretraining can reduce worst-case errors that occur with traditional supervised learning. Moreover, our observation that SSL percent errors consistently either matched or were lower than supervised percent errors indicates SSL pretraining also increases the robustness of models in label-limited scenarios.

However, we also observed that optimally trained models sometimes had similar or even worse performance than fully-supervised models for CT tissue cross-sectional area, CT mean HU value, and MRI T2 relaxation time in scenarios with 100% or 50% labeled data. This observation suggests that SSL does not have much benefit when the labeled dataset is large. In such cases, it may be more efficient to simply train a fully-supervised model, rather than spend additional time pretraining with unlabeled data.

When training with 5% labeled data for all MRI classes and muscle on CT, our optimal pretraining strategy improved Dice scores by over 0.05, compared to fully-supervised learning. In such cases, the Dice score for fully-supervised learning was 0.8 or lower, which suggests a critical performance threshold where inpainting-based SSL can improve segmentation performance over supervised learning. SSL may be beneficial in these cases because the models still have the capacity to learn more meaningful representations, compared to models with Dice scores over 0.8 that may already be saturated in their capacity to represent the underlying image.

Importantly, it should be noted that the improvement in segmentation performance with SSL pretraining in label-limited scenarios is on the similar order as prior advances that used complex DL architectures and training strategies [34,35,36]. Comparatively, our proposed SSL training paradigm offers an easy-to-use framework for improving model performance for both MRI and CT without requiring large and difficult to train DL models. Moreover, since we have already investigated different implementation design choices and experimentally determined the best ones, our proposed training paradigm will provide researchers with an implementation of inpainting-based SSL for their own work, without requiring them to spend resources/compute investigating these design choices again. This is especially important as we have shown that simply performing inpainting-based pretraining on the same data that is ordinarily only used for supervised learning improves segmentation accuracy compared to supervised learning only.

### Study Limitations

There were a few limitations with this study. Although we investigated two different methods for selecting which pretrained weights to transfer, we did not conduct a systematic study across all possible choices due to computational constraints that made searching over the large search space too inefficient. We also leave other SSL strategies such as contrastive learning to future studies since it requires systematic evaluation of augmentations and sampling strategies. Furthermore, when we investigated the impact of unlabeled data extents on downstream segmentation performance, we did not pretrain our SSL models with equal extents of unlabeled MRI and CT data since we maximized the amount of available MRI data. In addition, our investigations in this work are limited to the U-Net architecture, though future work can explore other powerful segmentation architectures. Finally, we did not experiment with other optimizers potentially better than the ADAM optimizer. Recent studies [37] have shown that there may be value in optimizers such as Stochastic Gradient Descent for better generalization in natural image classification and that there is potential trade off while choosing different optimizers. We leave the systematic investigation of this issue on medical imaging data for future follow up work.

## 5. Conclusions

In this work, we investigated how inpainting-based SSL improves MRI and CT segmentation compared to fully-supervised learning, especially in label-limited regimes. We presented an optimized training strategy and open-source implementation for performing such pretraining. We describe the impact of pretraining task optimization and the relationship between the sizes of labeled and unlabeled training datasets. Our proposed approach for pretraining for improving segmentation performance that does not require additional manual annotation, complex model architectures, or model training techniques. 

## Figures and Tables

**Figure 1 bioengineering-10-00207-f001:**
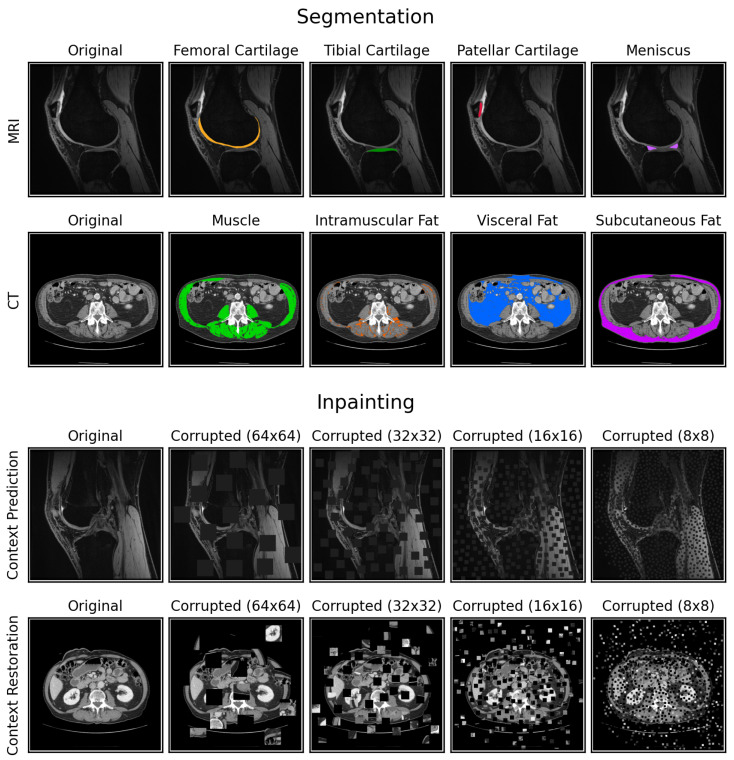
Example ground truth segmentations for the MRI and CT datasets (both with dimensions 512 × 512), and example image corruptions for context prediction (zero-ing image patches) and context restoration (swapping image patches). Since image corruption happens after normalization, the zero-ed out image patches for context prediction were actually replaced with the mean of the image. The “Inpainting” section depicts image corruptions with four different patch sizes: 64 × 64, 32 × 32, 16 × 16, and 8 × 8. The locations of these patches were determined using Poisson-disc sampling to prevent randomly overlapping patches.

**Figure 2 bioengineering-10-00207-f002:**
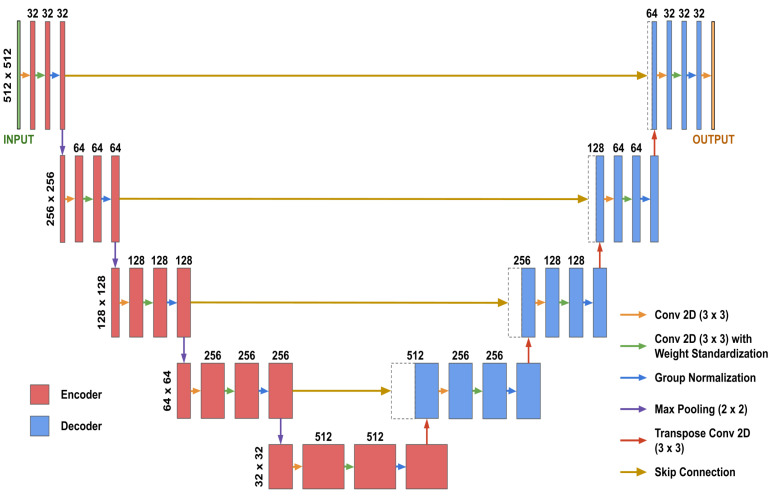
The U-Net architecture used for both inpainting and segmentation, which includes layers grouped into three categories: the “encoder” (in red), the “decoder” (in blue), and the “post-processing” layer (the final convolutional layer). Each dotted rectangular box represents a feature map from the encoder that was concatenated to the first feature map in the decoder at the same level.

**Figure 3 bioengineering-10-00207-f003:**
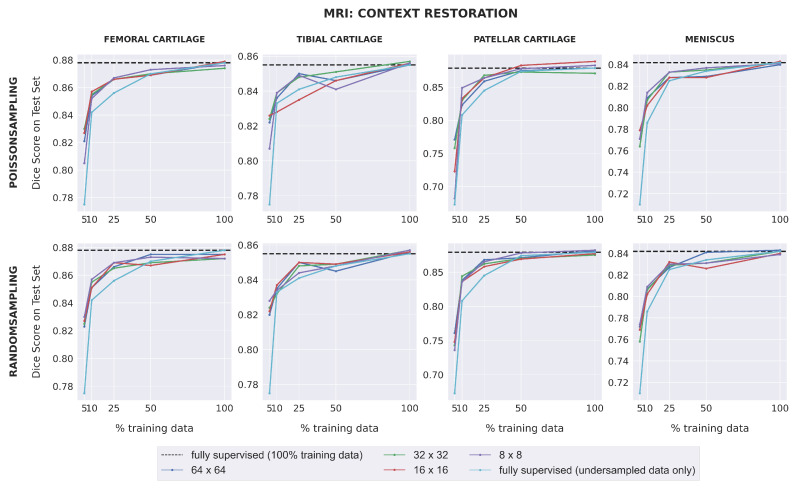
The downstream segmentation performance on the MRI dataset for the Context Restoration pretext task as measured by the Dice score for every combination of patch size and sampling method used during pretraining, evaluated in five different scenarios of training data availability. In each scenario, every model is trained for segmentation using one of the five different subsets of training data as described in Section 2.1.1. The black dotted line in each plot indicates the performance of a fully-supervised model trained using all available training images. The light blue curve indicates the performance of a fully-supervised model when trained using each of the five different subsets of training data. Similar plots for the Context Prediction pretext task are given in Appendix C.

**Figure 4 bioengineering-10-00207-f004:**
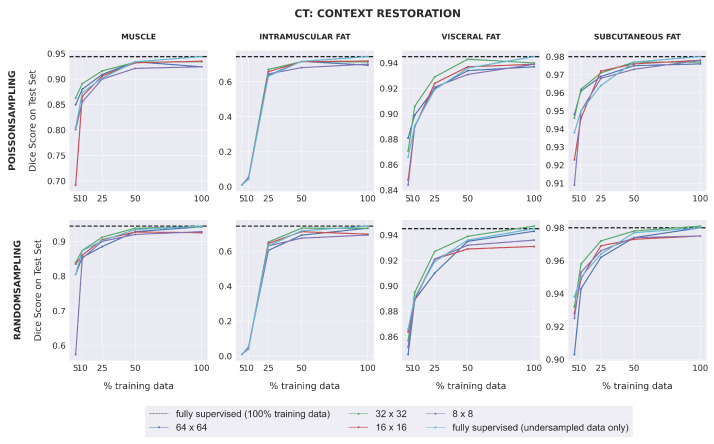
The downstream segmentation performance on the CT dataset for the Context Restoration pretext task as measured by the Dice score for every combination of patch size and sampling method used during pretraining, evaluated in five different scenarios of training data availability. In each scenario, every model is trained for segmentation using one of the five different subsets of training data as described in Section 2.1.2. The black dotted line in each plot indicates the performance of a fully-supervised model trained using all available training images. The light blue curve indicates the performance of a fully-supervised model when trained using each of the five different subsets of training data. Similar plots for the Context Prediction pretext task are given in Appendix C.

**Figure 5 bioengineering-10-00207-f005:**
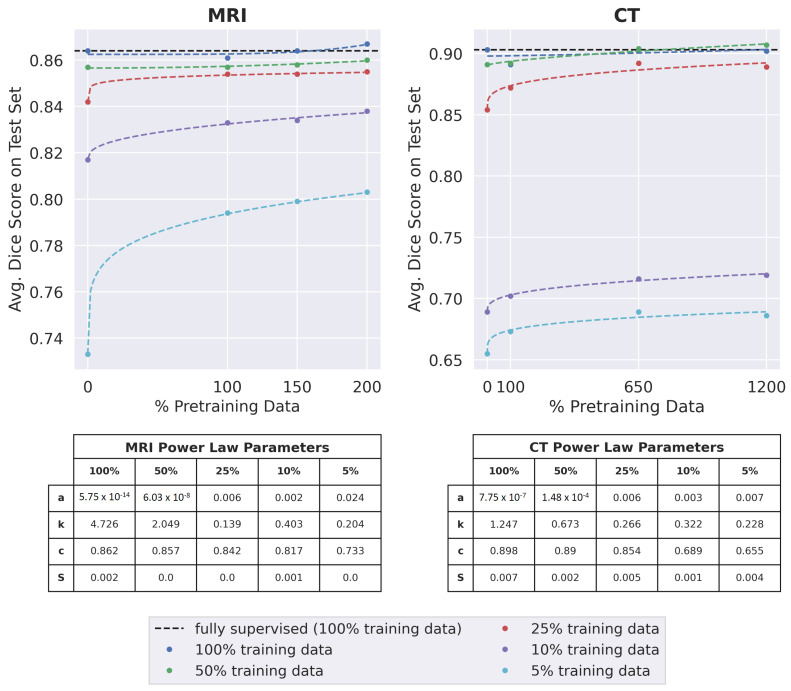
The downstream segmentation performance of the optimally trained model when pretrained with different amounts of pretraining data and fine-tuned using each of the five training data subsets. 100% pretraining data refers to the regular training set for each dataset. The data point for 0% pretraining data is the performance of a fully-supervised model. The black dotted line indicates the performance of a fully-supervised model trained on all available training data for the appropriate dataset. The other dotted lines are the best-fit curves for each of the training data subsets, modeled as a power-law relationship of the form $y=ax^{k}+c$. The values of *a*, *k*, *c*, and the Residual Standard Error (*S*) for the best-fit curves are displayed in the two tables.

**Figure 6 bioengineering-10-00207-f006:**
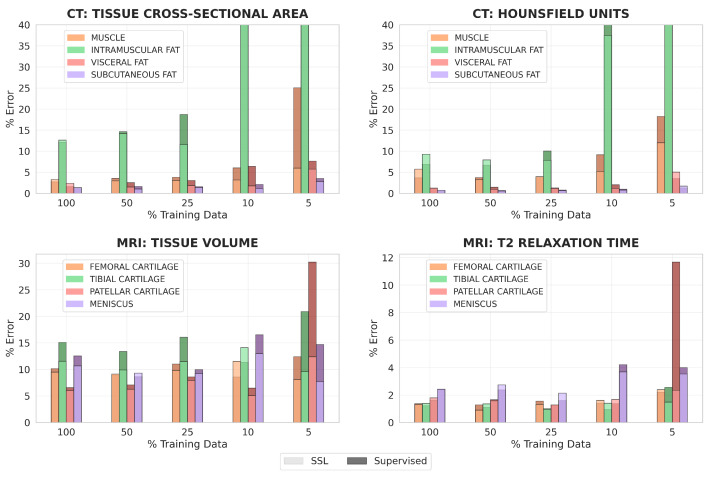
A comparison of the percent error in calculating clinical metrics for the MRI and CT datasets between when the tissue segmentations are generated by fully-supervised models and when the tissue segmentations are generated by optimally trained models, pretrained using 200% data for MRI and 1200% data for CT. Each bar represents the median percent error across the test set for a particular tissue, clinical metric, and label regime. The percent error in the calculation of tissue cross-sectional area and mean HU for intramuscular fat extends beyond the limits of the y-axis when 10% and 5% labeled training data for segmentation is used.

**Figure 7 bioengineering-10-00207-f007:**
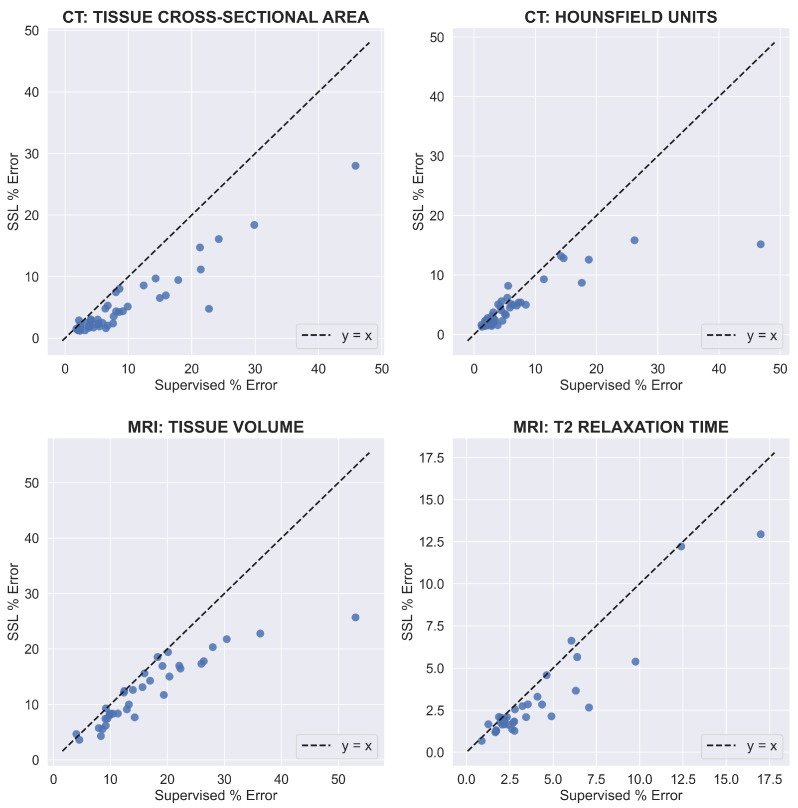
The relationship between the percent error when using supervised learning and the percent error when using SSL. Each blue point represents an image in the test set for the appropriate dataset. The percent error was averaged over all classes and label-limited scenarios. For CT, the intramuscular fat was excluded to prevent large percent error values. For MRI T2 relaxation time, one point with a high percent error for supervised learning was excluded to reduce the range of the x-axis.

**Table 1 bioengineering-10-00207-t001:** Demographics of the subjects included in this study. Age is shown as mean ± standard deviation. For the CT dataset, one subject did not have age information and four subjects did not have gender information.

MRI			
**Split**	**Gender**	**# Volumes (# Slices)**	**Age (Range)**
Train	Male	46 (7360)	44.7 ± 17.7 (17–75)
	Female	40 (6400)	42.9 ± 18.5 (16–87)
	Total	86 (13760)	43.9 ± 18.1 (16–87)
Validation	Male	18 (2880)	37.3 ± 16.8 (18–68)
	Female	15 (2400)	53.2 ± 14.9 (18–79)
	Total	33 (5280)	44.5 ± 17.8 (18–79)
Test	Male	26 (4156)	37.9 ± 14.9 (18–71)
	Female	10 (1584)	53.0 ± 11.9 (31–73)
	Total	36 (5740)	42.1 ± 15.6 (18–73)
Unlabeled	Male	37 (5446)	38.1 ± 16.9 (15–77)
	Female	49 (6686)	52.1 ± 18.5 (14–97)
	Total	86 (12132)	46.1 ± 19.1 (14–97)
**CT**			
**Split**	**Gender**	**# Slices**	**Age (Range)**
Train	Male	362	68.2 ± 11.4 (20–97)
	Female	343	71.1 ± 10.5 (18–95)
	Total	709	69.6 ± 11.1 (18–97)
Validation	Male	63	69.1 ± 9.5 (32–83)
	Female	69	71.0 ± 11.0 (32–89)
	Total	133	70.1 ± 10.4 (32–89)
Test	Male	18	70.6 ± 11.9 (47–92)
	Female	26	73.1 ± 11.7 (44–93)
	Total	44	72.1 ± 11.9 (44–93)
Unlabeled	Male	3167	51.5 ± 17.1 (18–101)
	Female	4632	51.6 ± 17.1 (18–100)
	Total	7799	51.6 ± 17.1 (18–101)

**Table 2 bioengineering-10-00207-t002:** Quantitative evaluation of inpainting for every combination of pretext task, patch size, and sampling method. All values are rounded to the nearest integer.

Pretext Task and Patch Size		MRI		CT	
		L2 Norm (Mean ± Std)		L2 Norm (Mean ± Std)	
**Pretext Task**	**Patch Size**	**Poisson-Disc**	**Random**	**Poisson-Disc**	**Random**
**Context** **Prediction**	64 × 64	94 ± 9	105 ± 9	123 ± 13	134 ± 18
	32 × 32	75 ± 8	81 ± 8	83 ± 9	112 ± 14
	16 × 16	61 ± 7	64 ± 7	66 ± 8	74 ± 10
	8 × 8	51 ± 6	52 ± 5	54 ± 7	57 ± 8
**Context** **Restoration**	64 × 64	96 ± 9	116 ± 11	142 ± 19	346 ± 158
	32 × 32	75 ± 8	84 ± 8	108 ± 12	127 ± 15
	16 × 16	62 ± 7	67 ± 7	86 ± 10	93 ± 12
	8 × 8	51 ± 5	56 ± 6	66 ± 8	80 ± 9

**Table 3 bioengineering-10-00207-t003:** Summary of the top five combinations of pretext tasks, patch sizes, and sampling methods for each dataset with the corresponding *p*-value for each combination, and sorted by *p*-value in ascending order. The bolded pretext task, patch size, and sampling method were chosen as the best combination of the three design choices.

MRI				
**Rank**	**Pretext Task**	**Patch Size**	**Sampling Method**	* **p** * **-Value**
1	Context Restoration	64 × 64	Random	1.64 × $10^{-18}$
2	Context Restoration	8 × 8	Random	1.89 × $10^{-18}$
3	**Context Restoration**	**32** × **32**	**Poisson-Disc**	1.05 × $10^{-17}$
4	Context Restoration	8 × 8	Poisson-Disc	4.03 × $10^{-17}$
5	Context Restoration	32 × 32	Random	9.38 × $10^{-16}$
**CT**				
**Rank**	**Pretext Task**	**Patch Size**	**Sampling Method**	* **p** * **-Value**
1	**Context Restoration**	**32** × **32**	**Poisson-Disc**	6.29 × $10^{-17}$
2	Context Restoration	64 × 64	Poisson-Disc	1.88 × $10^{-9}$
3	Context Restoration	32 × 32	Random	4.12 × $10^{-7}$
4	Context Prediction	64 × 64	Poisson-Disc	0.1
5	Context Prediction	64 × 64	Random	0.66

## Data Availability

All MRI data can be accessed via the publicly-shared SKM-TEA data repository (https://github.com/StanfordMIMI/skm-tea), CT data can be made available via request to the authors in a manner compliant with IRB approvals and human subjects research.

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
