# Peer review of "Improving Data-Efficiency and Robustness of Medical Imaging Segmentation Using Inpainting-Based Self-Supervised Learning"

_bioengineering, 2023, doi:10.3390/bioengineering10020207_

Round 1

Reviewer 1 Report

This paper mainly evaluates the training methodology and efficacy of two inpainting-based pretext tasks of context prediction and context restoration for medical image segmentation using self-supervised learning (SSL). The experimental results demonstrate that SSL pretraining using inpainting-based pretext tasks can help increase the robustness of models in label-limited scenarios and reduce worst-case errors that occur with supervised learning. The overall structure is clear. However, there are some concerns as follows:

(1) It is recommended to provide more discussions about the advantages of the presented optimized training strategy.

(2) A discussion of the limitations of the proposed solution is currently missing. Besides,  it is not clear in which cases the proposed method will fail.  

(3) Some related works are missing, including 1) Self-supervised Multi-modal Hybrid Fusion Network for Brain Tumor Segmentation, IEEE JBHI 2022; 2) Inf-net: Automatic covid-19 lung infection segmentation from ct images, IEEE TMI 2020.

Reviewer 2 Report

In this paper, the authors evaluated the efficacy of two inpainting-based pretext tasks, including context prediction and context restoration, for medical image segmentation using self-supervised learning (SSL). The authors tested the initial learning rate of fine-tuning, the image patch size and sampling methods of the image patch locations in image corruption in MRI and CT datasets. The structure of the paper is well organized and the content is comprehensive. However, no new network structure was designed, and no new pretext task was used, the authors only tested the existing methods, so the innovation of the paper is not obvious. Here are some suggestions for the authors to improve the paper.

1.      The initial learning rate is related to the optimization algorithm and network structure. The authors selected ADAM for optimization and u-net for image segmentation. Adam is an adoptive learning rate optimizer. It sometimes works, and sometimes doesn't. So I suggest the authors try some other optimizer, e.g. SGD+Nesterov momentum. It is more meaningful than just testing the initial learning rates.

2.      I think the authors should think about the significance of sampling techniques in image corruption. In my opinion, SSL uses pretext tasks to mine the supervision information from large-scale unlabeled data. So it has little to do with the different sampling methods if there are enough data, which was verified by the experimental results in the paper.

3.      The experiment showed some obvious conclusions. For example, with the increase of the pretraining data, we can get better segmentation results. I hope the authors could find out more meaningful conclusions from the experiments.

4.      Due to the particularity of medical images, it is difficult to build large-scale training datasets of medical images, so, SSL is very meaningful in medical image processing. It is suggested that the authors design or improve pretext tasks based on the characteristics of MRI and CT images to solve the problems in pre-training, e.g. hard example mining, or saturation of the learning capacity of a model, as the authors said.

Round 2

Reviewer 2 Report

In this paper, the authors evaluated the efficacy of two inpainting-based pretext tasks for medical image segmentation using self-supervised learning (SSL). But the authors only tested the existing methods, no new network structures or new pretext tasks. Moreover, some evaluation parameters are unnecessary, and there is no specific evaluation method for MRI and CT datasets. In the authors' response, the authors did not respond well to the reviewer's comments and did not add enough experiments. The paper lacks innovation. I don't think the paper is suitable for publication in Bioengineering, which is a kind of engineering Journal.